# Endosonography-Guided Versus Percutaneous Gallbladder Drainage Versus Cholecystectomy in Fragile Patients with Acute Cholecystitis—A High-Volume Center Study

**DOI:** 10.3390/medicina58111647

**Published:** 2022-11-14

**Authors:** Hayato Kurihara, Francesca M. Bunino, Alessandro Fugazza, Enrico Marrano, Giulia Mauri, Martina Ceolin, Ezio Lanza, Matteo Colombo, Antonio Facciorusso, Alessandro Repici, Andrea Anderloni

**Affiliations:** 1Emergency Surgery Unit, Fondazione IRCCS Ca’ Granda Ospedale Maggiore Policlinico, 20122 Milan, Italy; 2Emergency Surgery and Trauma Section, Department of Surgery, IRCCS—Humanitas Research Hospital, 20089 Rozzano, Italy; 3Department of Biomedical Sciences, Humanitas University, 20090 Pieve Emanuele, Italy; 4Division of Gastroenterology and Digestive Endoscopy, Department of Gastroenterology, IRCCS—Humanitas Research Hospital, 20089 Rozzano, Italy; 5General Surgery, Hospital Universitari Parc Taulí, 08208 Sabadell, Spain; 6Department of Diagnostic and Interventional Radiology, IRCCS—Humanitas Research Hospital, 20089 Rozzano, Italy; 7Department of Medical and Surgical Sciences, Gastroenterology Unit, University of Foggia, 71122 Foggia, Italy; 8Gastroenterology and Digestive Endoscopy Unit, Fondazione IRCCS Policlinico San Matteo, 27100 Pavia, Italy

**Keywords:** cholecystitis, percutaneous gallbladder drainage, endosonography-guided gallbladder drainage, cholecystectomy, frailty, ACS score

## Abstract

*Background and Objectives:* Acute cholecystitis is a frequent cause of admission to the emergency department, especially in old and frail patients. Percutaneous drainage (PT-GBD) and endosonographic guided drainage (EUS-GBD) could be an alternative option for relieving symptoms or act as a definitive treatment instead of a laparoscopic or open cholecystectomy (LC, OC). The aim of the present study was to compare different treatment groups. *Materials and Methods:* This is a five-year monocentric retrospective study including patients ≥65 years old who underwent an urgent operative procedure. A descriptive analysis was conducted comparing all treatment groups. A propensity score was estimated based on the ACS score, incorporated into a predictive model, and tested by recursive partitioning analysis. *Results:* 163 patients were included: 106 underwent a cholecystectomy (81 laparoscopic (LC) and 25 Open (OC)), 33 a PT-GBD and 21 EUS-GBD. The sample was categorized into three prognostic groups according to the adverse event occurrence rate. All patients treated with EUS-GBD or LC resulted in the low risk group, and the adverse event rate (AE) was 10/96 (10.4%). The AE was 4/28 (14.2%) and 21/36 (58.3%) in the middle- and high-risk groups respectively (*p* < 0.001). These groups included all the patients who underwent an OC or a PT-GBD. The PT-GBD group had a lower clinical success rate (55.5%) and higher RR (16,6%) when compared with other groups. *Conclusions:* Surgery still represents the gold standard for AC treatment. Nevertheless, EUS-GBD is a good alternative to PT-GBD in terms of clinical success, RR and AEs in all kinds of patients.

## 1. Introduction

Acute cholecystitis (AC) is one of the common differential diagnoses for patients presenting with abdominal pain in the emergency department. The risk of developing symptomatic episodes of biliary colic and cholecystitis is higher in the older population. Early cholecystectomy is the gold standard of treatment for AC [1,2]. However, in elderly, polymorbid and high-risk patients, alternative therapies should be considered such as percutaneous gallbladder drainage (PT-GBD) or endoscopic ultrasound-guided gallbladder drainage (EUS-GBD) [3]. Several studies reported comparisons between surgery and alternatives techniques such as the CHOCOLATE trial [4] and more recently Teoh et al. [5,6]. Current guidelines do not provide definitive recommendations on how to manage high-risk patients with AC who would not be considered for surgery [2,7]. Due to the lack of evidence, we decided to investigate our series by comparing all treatment populations for AC.

## 2. Materials and Methods

Data were collected from a prospectively maintained database at the Humanitas Research Hospital, IRCCS, Rozzano, Italy (with approval of the Ethical Committee of the IRCSS Humanitas Research Hospital of Rozzano – Milan; NCT02855151 (20/01/2016)). All patients ≥65 years old accepted at the emergency department with AC who underwent an urgent operative procedure (PT-GBD, EUS-GBD, LC or OC) between January 2015 and December 2020 were included. Data included: sociodemographic and preoperative data and medical history (previous episodes of AC, nonoperative management (NOM) failure, previous ERCP, anticoagulant medications, laboratory tests). The severity of the disease was calculated according to the latest Tokyo guidelines (TG18) that state AC into three severity grades. Each AC case was evaluated and its severity scored according to TG18 [2]. The American College of Surgeons National Surgical Quality Improvement Program Surgical Risk Calculator (ACS NSQIP score SRC) was used to estimate the serious complication risk and the death risk compared with the average risk of the procedure [8]. We defined it as ACS score. The Clinical Frailty Scale Score (CFSS) was used to define the fragility level [9]. The failure of nonoperative management (NOM) was reported even if patients were included independent of the fact that a conservative treatment had been previously attempted.

The therapeutic strategy was decided by a multidisciplinary group composed of surgeons, endoscopists, and radiologists choosing for each patient the best procedure according to the above-mentioned score and the best practice habits. If surgery was the choice, the laparoscopic or open approach was chosen by the attending surgeon considering the patient physiology, the TG18 score, the AAST score, and lastly the laparoscopy expertise (although at our center, all the senior surgeons have good laparoscopy expertise, and if a difficult LC was performed by a surgery resident, a further laparoscopic procedure was attempted by the senior surgeon before converting to open surgery).

Ongoing anticoagulant therapy was considered a risk factor for complications.

Postoperative variables were classified as: adverse events (AE), technical success (TS), clinical success (CS), need for rescue procedure, kind of rescue procedure, recurrence rate (RR), length of hospital stay, 90-day mortality.

### 2.1. Outcomes Definitions

The first outcome was technical success (TS), and it was defined differently for each technique: the ability to drain the gallbladder with the placement of a lumen-apposing metal stent (LAMS) in the EUS-GBD group; performing a complete cholecystectomy, laparoscopic or open; or placing a percutaneous catheter in the PT-GBD group.

The second outcome was the clinical success, considered the improvement of clinical symptoms and laboratory tests within the first week after the procedure [5,10].

The third outcome was the recurrence rate (RR), considered a further AC episode within one year from the first procedure or the immediate need of a rescue procedure due to a lack of clinical improvement.

As a last but major outcome, we considered the adverse event (AE) rate and the serious adverse event (SAE) rate. We registered AEs according to the Clavien Dindo scale (CD) considering them severe if CD ≥3 [11]. We considered infective complications all cases that required antibiotics, both oral and intravenous; we defined septic shock according to the last 2016 Sepsis-3 consensus [12].

Patients were followed up through both clinical visits and phone calls.

### 2.2. Procedures

LC or OC: In case of a laparoscopic approach, the procedure was performed using a four-trocar technique [2], while the open technique was performed through subcostal incision [13]. The critical view of safety [7] was always required before the cystic duct and cystic artery were transected after clipping them. The gallbladder was subsequently removed from its bed by electrocautery. Preoperative antibiotic prophylaxis was administered (2 gr. cefazolin e.v. or piperacilline-tazobactam in more severe grades of AC) [2].

PT-GBD: A percutaneous catheter was placed under image guidance (US or CT scan). The puncture was performed transhepatic or transperitoneal, and the pigtail catheter (8–10 Fr) was placed either with one-step using a trocar or with the Seldinger technique. The drain would stay in place for almost three weeks and required an antegrade cholangiography before its removal [4,5,6,14].

EUS-GBD: This procedure provides a connection between the gastrointestinal lumen and the gallbladder by placing a LAMS. Under EUS guidance, the gallbladder was studied and drained from either the stomach or duodenum. It consisted in accessing the gallbladder directly by puncturing it with the device on the pure-cut setting, followed by deployment of a 8–8 mm or a 10–10 mm stent without any exchange of devices [10,15].

### 2.3. Statistical Analysis

Patient characteristics were summarized using median and interquartile range (IQR) for continuous variables and absolute frequencies and percentages for categorical ones. Variables were compared with chi square test in the case of categorical variables and ANOVA in the case of continuous ones. To assess the prognostic role of the baseline variables on AE rate and to rank the relevance of each predictor variable, we estimated random-forest variable importance measures through the permutation of variable values. To control for pretreatment imbalances on observed variables and better estimate the causal effects of interventions, a propensity score was estimated with a generalized boosted model on the following variables: age, sex, ACS, previous episode of AC, antithrombotic therapy, WBC, and CRP. The propensity score was then incorporated into the predictive model through inverse probability weighting [16,17,18]. The predictor variables were then tested by recursive partitioning analysis (RPA) [19]. Once risk groups were identified in the model, they were compared to determine whether sufficient divergences in terms of adverse event rate and severe adverse event rate were present across identified risk classes, by means of chi square test. The model was then internally validated using bootstrap resampling and the performance validation was expressed through the area under the receiver-operating characteristic curve (AUC, or c-statistic) [20]. Calibration was assessed by means of a plot showing the correlation between the mean predicted AE probability versus the mean observed AE rate in deciles of patients with increasing values of the predicted probability. Differences between predicted probability and observed AE rate were assessed by the Hosmer–Lemeshow test [21]. All statistical tests were 2-tailed, and differences were considered significant at a *p* value < 0.05. The statistical analysis was run using the rpart and performance packages in R Statistical Software 3.0.2 (Foundation for Statistical Computing, Vienna, Austria).

## 3. Results

In this study, 163 patients were included. Surgery was performed in 106 patients (81 LC, 25 OC), PT-GBD on 33 cases and EUS-GBD on 21 cases. Among the group, 90 patients (56%) were males, with significant difference in sex distribution amongst groups. Median age was 77 (IQR 71–82) (*p* < 0.001). Patients who underwent surgery were younger. The same was for the ACS score distribution: it was higher in the PT-GBD group and in the OC group (*p* < 0.001). The width range was 1.00–8.00, and the median CFSS was 3, but it was higher in the PT-GBD groups, and its difference was statistically significant (*p* < 0.001).

Considering the pretreatment laboratory values, a significant difference was found amongst groups for WBC count (*p* = 0.002) and CRP (*p* = 0.004). Differences in INR values were not significant, but our study outlined relevant differences in distribution among groups. A statistically significant distribution was observed for the anticoagulant therapy: over half (57%) of the patients who underwent EUS-GBD were on active antithrombotic therapy (*p* = 0.02) and the therapy wasn’t suspended before the procedure.

The failure of eventual nonoperative management was reported with a statistically significant difference among groups: patients from PT-GBD group resulted in a higher rate of NOM failure (51.5%).

The preoperative baseline characteristics of patients are outlined in Table 1.

A detailed list of patient outcomes is reported in Table 2.

TS was reached in 100% of cases except in the EUS-GBD group, where in one case, the device positioning was not feasible due to the migration of the device between the gallbladder lumen and the stomach. A rescue surgery was performed with the repair of a microscopic antral wall defect.

In the surgical group, four patients were converted from LC to OC, but all the surgeries were completed successfully without cases of incomplete cholecystectomy. All the AC that required a change of surgical plan had an AAST score of more than 3 and a TG18 score of 2 or 3. Furthermore, during surgery they badly tolerated pneumoperitoneum, and the surgical team finally decide to convert. One of them was a severely ill patient with a frailty score of 8 who failed NOM and was surgically treated. He finally died during intensive care hospitalization due to his extremely poor physiological status.

Overall CS rate was 124/163 (78%), and RR was 15/163 (9%). The overall AE rate was 41/163 (26%). Of all AEs (41), only 17 were graded as severe and required invasive treatment. We registered five cases of PT-GBD displacement managed by the repositioning of the drain.

Six patients experienced a biliary leak (five in the OC group and one PT-GBD). Of these, the one from the PT-GBD group was due to a Luschka duct and was surgically treated, while the others resolved by leaving the drains in place. In five cases (all from the PT-GBD group), patients experienced severe septic shock. In one of them, a rescue EUS-GBD was performed, and the others patients were surgically treated.

No bleeding complications occurred, and the EUS-GBD series patients on active anticoagulant therapy were not burdened by a higher number of AEs. Infective complications occurred in 10 patients; 3 cholangitis and 7 severe septic shocks: 4 from the PT-GBD group, 1 from the EUS-GBD group, 2 from OC group and 1 from the LC. The 4 patients from the PT-GBD group, underwent a rescue procedure: 2 surgeries, 1 repositioning of percutaneous drainage and 1 EUS-GBD. The endoscopically treated patient and one of the two surgically treated patients died due to medical complications (pneumonia and cardiac failure).

In the results, 90-day mortality was 5/163 (3%): 3 patients from the PT-GBD group and 2 from the surgical group. Of the PT-GBD group, two patients died after a rescue cholecystectomy due to a serious septic status. The others died because of medical complications. One case of cholangitis was reported after surgery, and it was treated with antibiotics.

Additional data about adverse events are presented in Appendix A.

Looking at the RR, 11 patients returned to the emergency department for a new AC episode. Of these, 7 were from the PT-GBD group, 3 from the EUS-GBD group. Patients from PT-GBD group were managed with EUS-GBD in 2 cases and LC in 2 cases, and the others were treated non-operatively with antibiotics. Furthermore, a case of cholangitis was described in the LC group causing readmission of the patient. It was managed with antibiotics with no complications.

Patients from the EUS-GBD groups who developed a recurrent AC episode were all treated non-operatively with good response.

The median LOS for PT-GBD patients was 15,26 days while it was 5 days for patients who underwent EUS-GBD, 4.61 for LC and 8.12 for OC. All the hospitalization was considered and not only the time after the procedure. Most patients who underwent PT-GBD were severely comorbid and frail, and they developed AC during the hospitalization for other issues.

### 3.1. Recursive Partitioning Analysis

As reported in Figure 1, random forest analysis found that greater importance in predicting AE occurrence was related to treatment used and ACS score. In fact, the greater permutation accuracy importance was observed in the treatment (EUS-GBD, PC-GBD, open surgery or VLS) and ACS score, followed by clinical frailty score and age. RPA of prognostic factors demonstrated that the risk of AE was stratified according to treatment used and ACS score, as in Figure 2. These factors split the series into three classes: low, intermediate, and high risk. Patients treated with EUS-GBD and patients with ACS score <17.4 treated with LC fall in the low-risk group (AE 10.4%, SAE 3.1%). Patients with ACS score <17.4 treated with open surgery or PT-GBD are classified at intermediate risk (AE 14.2%, SAE 7.1%), whereas patients with ACS score >17.4 treated with surgical (either LC or open) or percutaneous approach were at high risk of AE (AE 58.3%, SAE 25%). Of note, patients in the high-risk group experienced also poorer efficacy outcomes, with a success rate of 55.5% (versus 87.5% and 89.2% of low and intermediate risk groups; *p* < 0.001) and RR of 16.6%, again significantly superiors (7.3% and 0%, respectively; *p* < 0.001). Higher values of mean decrease in accuracy importance indicate variables that are more important to the classification.

### 3.2. Performance of the Model and Validation

The model showed a c-index of 74.3% (95% CI: 71.1%-78.2%; Figure 3) and a proper calibration (Hosmer–Lemeshow *p* value = 0.58 and mean error rate 3.3%) as shown by the calibration plot (Wald test for calibration slope *p* = 0.55; Figure 4). The internal validation of the model based on a bootstrap method (250 repetitions), showed a c-index of 72.5% (95% CI 70.3%–75.4%) (Figure 3) thus excluding the risk of overfitting.

## 4. Discussion

The aim of the current study was to compare outcomes of EUS-GBD, PT-GBD and surgery (both Laparoscopic and Open approach) considering and comparing technical and clinical success, complications rate and recurrence [13,14,22,23,24,25]. In the last few years, an increasing interest for alternative treatments of AC in frail patients was registered and many studies focused on this topic [10,14,22,26,27,28,29].

From the baseline characteristics analysis, a different distribution amongst frailty and age was outlined; patients treated with PT-GBD were older than patients submitted to surgery and had a higher ACS score. CFS score was calculated for all patients and it outlined, at first, an extremely heterogenous distribution of fragility among the population, second, that patients undergoing PT-GBD resulted being frailer than others. Such a difference highlight, in our opinion, a good evaluation and treatment choice. In fact, these PT-GBD patients would have been expected to have a poor surgery response due to their frailty level. It must be noticed that patients undergoing EUS-GBD had a 2-point-lower CFSS. This is probably due to the retrospective nature of the current study and the short-time experience in EUS-GBD field. Patients with AC, candidate to an invasive procedure, were evaluated by surgeon, endoscopist and radiologist and frail patients should have been avoided from surgery and addressed to PT vs EUS guided drainage. However, the short-time expertise could have compelled the multi-disciplinary team to avoid EUS-GBD if not completely safe. In the end, the different distribution of treatments reflects the clinical practice where surgical treatment continues to be considered the gold standard also thanks to increased quality of anesthesiologic assistance [3,7]. The AE rate for the surgical branch did not result in a higher severe complication rate when compared to PT-GBD. The interim analysis of the CHOCOLATE trial recently demonstrated that patients who undergo surgery have a lower rate of major complications (12%) when compared with PT-GBD (44% of AE) [4]. In our series, 16% of the LC group had AEs vs 39% of the PT-GBD group [4]. A higher number of AEs was observed in the OC group (52%). Revising the OC series many cases of AC with high ASST and TG18 score were observed. Moreover, the ACS score was usually higher in patients undergoing an open approach, either due to the inner risk of the procedure and to the physio pathological state of the patient. The AE rate was quite high suggesting that these patients could have been good candidates to EUS-GBD instead of a surgical approach.

We choose the ACS as score considering its widespread use in the surgical community [8]. Patients with an ACS >17,4 had higher AE rate; all the patients treated with PT-GBD were part of this group (high risk of AE). On the other hand, patients who underwent LC had an ACS <17,4 and a lower AE rate. Considering this finding we may assume that the therapeutic strategy was correct and based on a valid risk estimation. Recently, Teoh et al., promoted EUS-GBD use, demonstrating that it may reduce the risk of gallstone-related adverse events. They demonstrated, using a propensity score matching, an AE rate of 13,3%, a RR of 10% with a clinical success of 93,3%. When compared to PT-GBD, they found that EUS-GBD significantly reduced 1-year AEs (10 (25.6%) vs. 31 (77.5%), *p* < 0.001) and 30-day AEs (5 (12.8%) vs. 19 (47.5%), *p* = 0.001). Based on our results we can support their findings (AE 10,4% vs. 14,2% in patients with ACS <17,4 and 58,3 in patients with ACS > 17,4) [5,6]. Similar were RR and CS (in our population respectively 7,3% and 87,5%). The main difference was the population selection, in Teoh’s series only patients excluded from surgery with an AC grade 2 and 3 were considered for EUS-GBD. In our study we considered all patients admitted with AC independently from the grade. This way EUS-GBD represents a real alternative to surgery in frail patients and not a second line procedure. In our center TS ranged almost 100% in all groups. However, multicentric study may be desirable to compare these procedures, also pondering the lack of experience, especially in EUS-GBD positioning [25,26,27,28,29].

CS was achieved in 64% of PT-GBDs, against 85% of both the EUS-GBD and LC groups. It may be assumed that EUS-GBD and LC are the best strategies, when looking at the first postoperative period. If we exclude the LC patients (RR = 0), the lower RR was observed in the EUS-GBD group (RR = 14% vs. 24% in PT-GBD). This could be due to the ease of displacement of the gallbladder drain or the higher risk of obstruction in a small drain lumen. 5 cases of dislodgment were registered in PT-GBD series (vs only one case in EUS-GBD group). This finding is in line with the current literature [22,23,24,25]. In a recent retrospective study the cholecystostomy tube dislodgment was registered 17,8% [30]. In Teoh’s series PT-GBD resulted in a higher rate of dislodgement (35%) while no EUS-GBD drain displaced. Another interesting finding is related to anticoagulant therapy did not influence the AE rate: the safety of this strategy has already been described in our center within a case series and it is confirmed in this setting [30]. Moreover, the device could theoretically stay in place until stones and sludge are completely drained and then removed or left in place lifelong if well tolerated by the patient. In our center no devices have been removed and until now any issue or complication or affection to the QoL have been reported. We are aware that a longer follow up could be desirable to better evaluate the efficiency of the device, recurrence rate and lastly the possible technical difficulty in performing future surgery. Our experience up to now provide results on the safety of the EUS-GBD procedure, on the clinical improvement and in shortening hospitalization. Future studies and the extension of the follow up on our patients will provide evidence on long-term efficacy.

## 5. Conclusions

Surgery for AC remains the gold standard treatment with good outcomes in recurrence rate, complications rate, safety and mortality. This paper evidence the emerging role of EUS-GBD, particularly in patients not fit for surgery. EUS-GBD could be considered a good alternative for treatment, long term results are promising even though still under study. This EUS procedure could represent an innovation, especially facing old and frail patients. Moreover, our results highlight the need for calculating scores and approaching patients in a multidisciplinary way even in an emergency setting.

## Figures and Tables

**Figure 1 medicina-58-01647-f001:**
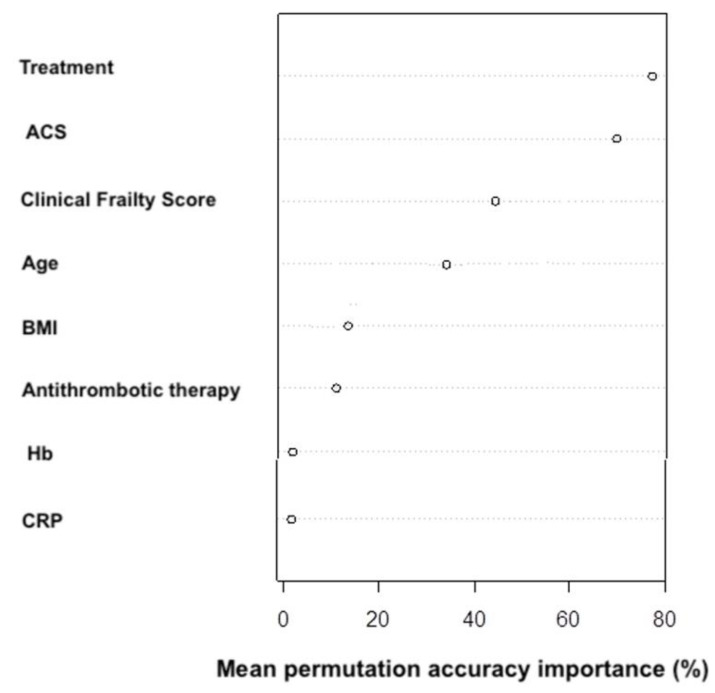
Variable importance estimated by permutation-based mean decreased accuracy importance considering adverse event rate as outcome.

**Figure 2 medicina-58-01647-f002:**
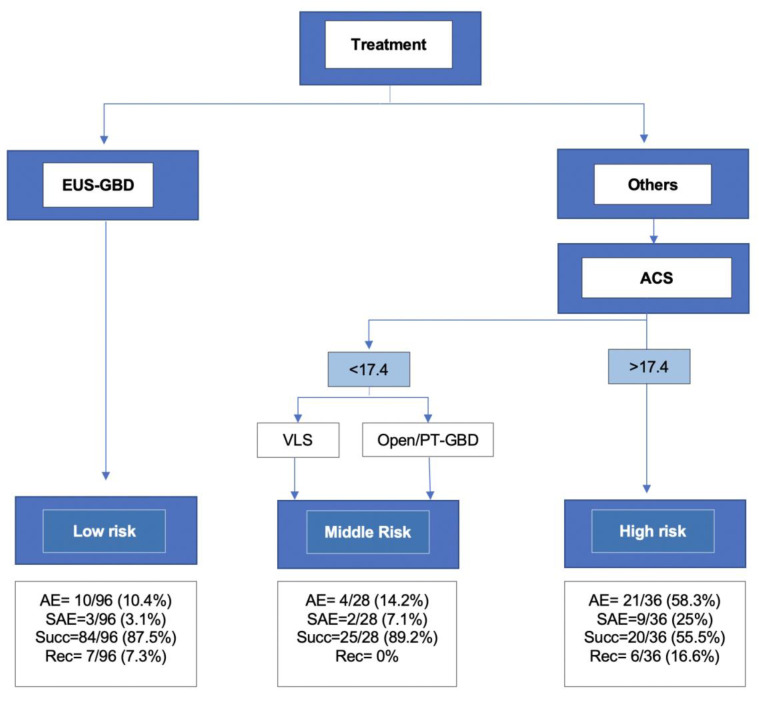
Recursive partitioning classification tree for adverse event occurrence. The terminal nodes categorized the study sample into three prognostic groups according to the adverse event occurrence rate. Adverse event rate was 10/96 (10.4%) in the low-risk group, 4/28 (14.2%) and 21/36 (58.3%) in the middle and high-risk groups, respectively (*p* < 0.001). Abbreviations: VLS: Videolaparoscopy, PT-GBD: Percutaneous Gallbladder drainage; EUS-GBD: Endoscopic ultrasound guided gallbladder drainage; ACS: American College of Surgeons risk score; AE: Adverse Event; SAE: Severe adverse event; Succ: Clinical success (CS); Rec: Recurrence (RR).

**Figure 3 medicina-58-01647-f003:**
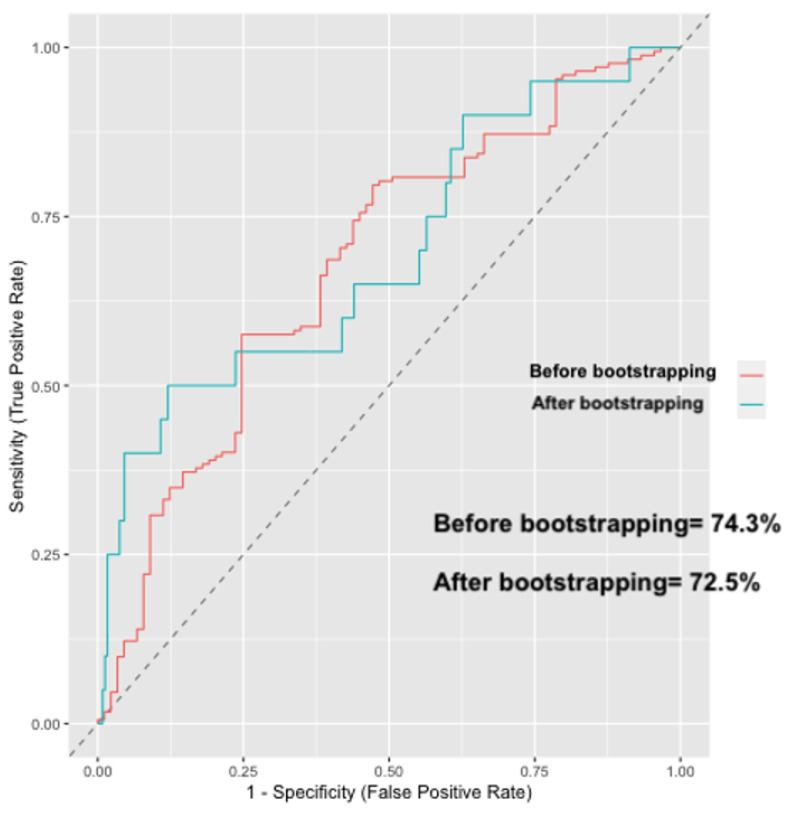
Receiver operating characteristic (ROC) curve and the corresponding area of the predictive model. Red line corresponded to the analysis before internal validation and green line corresponded to the analysis after bootstrapping-based internal validation.

**Figure 4 medicina-58-01647-f004:**
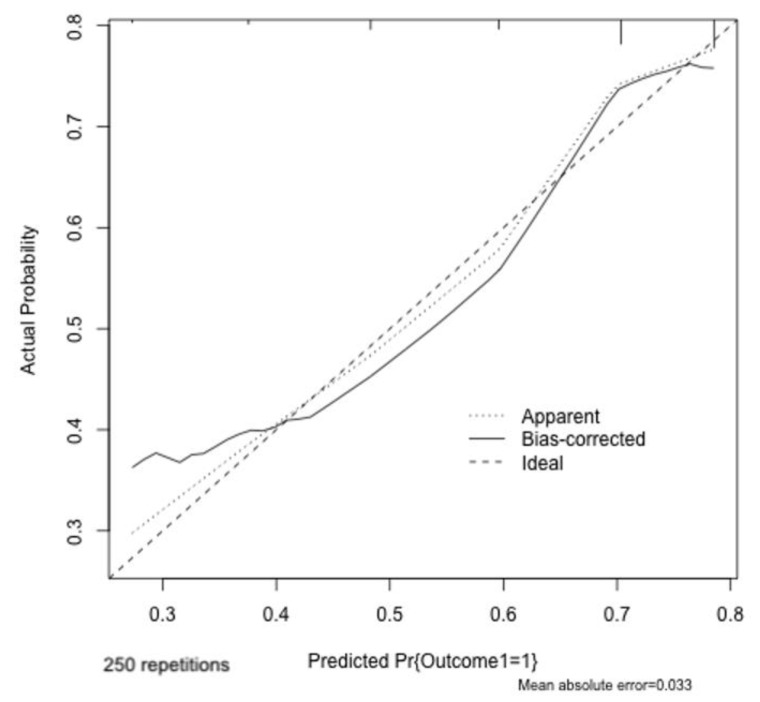
Calibration plot. Smoothed (loess) calibration plots reporting increasing predicted probability of adverse events by the assessed model. The diagonal line indicates the ideal line of perfect correspondence of predicted to observed adverse event rate. Mean absolute error rate was 3.3%.

**Table 1 medicina-58-01647-t001:** Baseline patients’ characteristics.

Variable	Total (*n* = 163)	PT-GBD (*n* = 33)	EUS-GBD (*n* = 21)	LC (*n* = 81)	OC (*n* = 25)	*p* Value
Age Median (IQR)	77 (71–82)	83 (75–87)	84 (81–89)	74 (70–79)	74 (69–80)	**<0.001**
Male No (%)	90 (56%)	12 (36%)	10 (48%)	51 (63%)	17 (68%)	**0.03**
BMI Median (IQR)	26.2 (24.2–29.1)	26.7 (23.6–29.6)	25.8 (21.6–28)	25.8 (24.3–25.8)	27.3 (25.2–31.2)	0.24
Clinical Fraility Scale Score (median [range])	3.00 [1.00, 8.00]	6.00 [2.00, 8.00]	4.00 [2.00, 8.00]	3.00 [1.00, 8.00]	3.00 [2.00, 7.00]	**<0.001**
ACS score Median (IQR)	6.45 (3.88–17.02)	17.6 (16.4,23)	6.1 (5.3–7.3)	4 (2.8–6)	17.5 (12.9–20.6)	**<0.001**
Non operative management (NOM) failure	36 (22.5)	17 (51.5)	4 (19.0)	11 (13.6)	4 (16.0)	**<0.001**
Previous AC No (%)	41 (26%)	12 (38%)	8 (38%)	17 (21%)	4 (16%)	0.10
Previous ERCP No (%)	17 (11%)	3 (9%)	6 (29%)	7 (9%)	1 (4%)	0.12
Anticoagulant therapy No (%)	47 (29%)	10 (30%)	12 (57%)	18 (22%)	7 (28%)	**0.02**
WBC No (×1000)	12 (8.5–16.5)	6.9 (0.7–16.1)	14.7 (7.5–17.1)	12.6 (10.4–17)	11.7 (10.7–14.4)	**0.02**
Hb [g/dl] Median (IQR)	12.9 (10.3–13.4)	NR	NR	13.1 (10.3–14.2)	12.4 (11–14.3)	0.11
CRP [mg/dl] Median (IQR)	11.5 (4–22.7)	15 (11.2–24)	6.15 (1.87–25)	8.32 (2.4–19.1)	13.5 (7.16–24.8)	**0.04**
Total Bilirubin [mg/dl] Median (IQR)	1.2 (0.73–2)	1.2 (0.7–2.4)	1.1 (0.62–1.53)	1.32 (0.88–2.08)	0.9 (0.68–1.87)	0.43
INR Median (IQR)	1.21 (1.1–1.37)	1.32 (1.21–1.69)	1.23 (1.1–1.36)	1.15 (1.07–1.31)	1.2 (1.13–1.29)	**0.008**

Continuous variables were reported as median values and interquartile range. Comparisons were performed with ANOVA for continuous variables and chi square test for categorical ones. Significances were reported in bold. Previous AC: previous episode of acute cholecystitis; ERCP: Endoscopic retrograde cholangiopancreatography; WBC: white blood cells; Hb: Hemoglobin; CPR: c-reactive protein; INR: International Normalized Ratio. Bold values are statistically significant.

**Table 2 medicina-58-01647-t002:** Postoperative Outcomes.

	Total (163 pts)	PT- GBD (*n* = 33)	EUS-GBD (*n* = 21)	LC (*n*= 81)	OC (*n* = 25)	
Technical Success (TS) No (%)	162 (99)	33 (100)	20 (95)	81 (100)	25 (100)	0.084
Clinical success (CS) No (%)	124 (78)	21 (64)	18 (85)	69 (85)	16 (64)	**0.02**
Recurrence Rate (RR) No (%)	11 (9)	7 (24)	3 (14)	0	1 (4)	**<0.001**
Adverse event rate No (%)	41 (26)	13 (9)	2 (10)	13 (16)	13 (52)	**<0.001**
Severe Adverse Event rate No (%)	17 (11)	9 (27)	0	2 (2)	4 (16)	**0.001**
**Type of adverse event**						
Drainage Displacement No (%)	6 (12)	5 (18)	0	-	-	NC
Sepsis/Septic Shock No (%)	8 (19)	4 (12)	1 (5)	2 (2)	2 (8)	NC
Cystic duct obstruction No (%)	1 (2)	1 (3)	-			NC
.cBiliary leak No (%)	6 (15)	1 * (3)	-	-	5 * (20)	NC
Cholangitis No (%)	1 (2)	-	-	-	1 (4)	NC
Medical complications (cardiological, respiratory, renal failure) No (%)	17 (41)	1 (3)	1 (5)	6 (7)	4 (16)	NC
Lenght of stay (day) (median [range])	6.00 [1.00, 167.69]	15.26 [5.00, 57.53]	5.00 [1.00, 167.69]	4.61 [1.12, 70.90]	8.12 [3.00, 53.82]	<0.001
90-Day mortality No (%)	5 (4)	3 (27)	-	2 (2)	0 (0)	**0.004**

* 1 secondary biliary duct. Bold values are statistically significant.

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
