# Peer review of "Endosonography-Guided Versus Percutaneous Gallbladder Drainage Versus Cholecystectomy in Fragile Patients with Acute Cholecystitis—A High-Volume Center Study"

_medicina, 2022, doi:10.3390/medicina58111647_

Round 1

Reviewer 1 Report

My congratulations to the authors for the topic of study. Very interesting study that looks for alternatives to surgery in elderly patients . This is a retrospective study comparing laparoscopic surgery, open surgery, percutaneous drainage, and endoscopic drainage in elderly patients with acute cholecystitis in a center with high experience in all this techniques. In daily clinical practice, we all treat patients with high surgical risk who develop acute cholecystitis. Since cholecystectomy is currently the standard, it is important to explore new alternatives for this group of patients with high surgical risk, since antibiotic treatment is sometimes not enough.

Reading the text, I have some methodological doubts that I am sure the authors will be able to clarify for a better understanding of the manuscript.

- The study does not describe the patients who were treated with antibiotics without the need for drainage. In patients with acute cholecystitis who do not undergo cholecystectomy, the alternative is usually management with antibiotics and perform percutaneous drainage or endoscopy only in the absence of response to antibiotics. Was this the case? Or was drainage performed without first assessing the response to the antibiotic? If so, it would be interesting to describe in a flow-chart which patients initially respond to antibiotic treatment to provide more information.  

- Has data on hospital stay been collected for each procedure? It would be interesting given that a longer hospital stay can have a negative influence on an elderly patient.

- Recurrence is described as another episode in one year and as the need for another procedure due to no clinical improvement. The recurrence is another episode in the following year, but wouldn't the need for another procedure due to no clinical improvement be a treatment failure? Would this not be included in the patients who do not have clinical success?  I do not see clearly the difference described by the authors between recurrence and clinical failure

- What were the criteria to choose whether the cholecystectomy was laparoscopic or open? What is the conversion rate of laparoscopic surgery? Which were initially chosen for the open surgery? In discussion it is commented that it could be due to hemodynamic instability but this is not reflected in the method and I think that is important to clarify this criteria. 

A table specifying the complications that occurred in each group and their classification according to a validated system such as the Clavien Dindo, would help to better understand the manuscript. Probably the authors cannot attach more tables but adding it as supplementary material could be considered if possible.  

- In the discussion, EUS-GBD is proposed as the first line of treatment, currently it is described as the second line after failure in medical treatment with antibiotics. If it is proposed as first line, in which case do the authors consider it to be an alternative to cholecystectomy? How have the patients been selected to be treated with one or another technique in this study? It is described in methods that have been decided by a multidisciplinary team, but if the authors affirm something as important as changing EUS-GBD to the first line of treatment, I think it would be important to define in which patients a cholecystectomy would be indicated and in which an EUS -GBD.

- What happened in patients undergoing cholecystectomy who were not clinically successful? Cholecystectomy usually resolves symptoms and corrects laboratory abnormalities caused by cholecystitis. Can the authors explain why these patients were not clinically successful?  

- The conclusions talk about using EUS-GBD as a bridge to surgery. Do the authors have experience in cholecystectomy after EUS-GBD? Does prior endoscopic drainage influence in the dificulty of a surgery? If the patient has undergone the EUS-GBD, I understand that it was due to high morbidity that contraindicated surgery, I understand that this morbidity will not change after admission, so I understand that the EUS-GBD is proposed as a definitive alternative to surgery given its low rate of adverse events and its low rate of recurrence of acute cholecystitis in patients with high morbidity in whom surgery would not be indicated. Is this what the authors propose?

I eagerly await the responses of the authors, which I am sure will help to better understand this interesting manuscript.

Author Response

First review response

My congratulations to the authors for the topic of study. Very interesting study that looks for alternatives to surgery in elderly patients. This is a retrospective study comparing laparoscopic surgery, open surgery, percutaneous drainage, and endoscopic drainage in elderly patients with acute cholecystitis in a center with high experience in all this techniques. In daily clinical practice, we all treat patients with high surgical risk who develop acute cholecystitis. Since cholecystectomy is currently the standard, it is important to explore new alternatives for this group of patients with high surgical risk, since antibiotic treatment is sometimes not enough.

Reading the text, I have some methodological doubts that I am sure the authors will be able to clarify for a better understanding of the manuscript.

- The study does not describe the patients who were treated with antibiotics without the need for drainage. In patients with acute cholecystitis who do not undergo cholecystectomy, the alternative is usually management with antibiotics and perform percutaneous drainage or endoscopy only in the absence of response to antibiotics. Was this the case? Or was drainage performed without first assessing the response to the antibiotic? If so, it would be interesting to describe in a flow-chart which patients initially respond to antibiotic treatment to provide more information. 

We collected retrospectively patients who underwent an invasive procedure, and we evaluated as preoperative variable the NOM failure (Non operative management failure) considering patients who did not improve with antibiotics. Nevertheless, we included all patients independently from the fact that a NOM was previously attempted. In the table 1 this value was not reported but I concur with the reviewer that it should be added (it was not in the final article due to words counts).

Overall

PT-GBD

EUS-GBD

LC

OC

p

Previous AC (%)

41 (25.8)

12 (37.5)

8 (38.1)

17 (21.0)

4 (16.0)

0,103

NOM failure (%)

36 (22.5)

17 (51.5)

4 (19.0)

11 (13.6)

4 (16.0)

<0.001

Moreover, the difference among the groups was statistically significant: Patients who underwent a PT-GBD were burdened in 51.5% by a failure of antibiotic therapy while only 13.6% of LC were previously treated with medical therapy and failed.

- Has data on hospital stay been collected for each procedure? It would be interesting given that a longer hospital stay can have a negative influence on an elderly patient.

Overall

PT-GBD

EUS-GBD

LC

OC

p

Lenght of stay (day) (median [range])

6.00 [1.00, 167.69]

15.26 [5.00, 57.53]

5.00 [1.00, 167.69]

4.61 [1.12, 70.90]

8.12 [3.00, 53.82]

<0.001

We concede that also in this case differences were outlined with statistical significance. The median LOS for PT-GBD patients was 15,26 days while it was 5 days for patients who underwent EUS-GBD, 4.61 for LC and 8.12 for OC. It must be taken into account that we considered all the hospitalization and not only the time after the procedure. Most patients who underwent PT-GBD were severely comorbid and frail, and they developed AC during the hospitalization for other issues.

Because of that this result is not explicative of the clinical success of the procedure. For further investigations might be desirable consider only the time after the procedure. Nevertheless, the variable “clinical success” partly takes into consideration the time after the procedure.

- Recurrence is described as another episode in one year and as the need for another procedure due to no clinical improvement. The recurrence is another episode in the following year, but wouldn't the need for another procedure due to no clinical improvement be a treatment failure? Would this not be included in the patients who do not have clinical success?  I do not see clearly the difference described by the authors between recurrence and clinical failure.

We are aware that the difference could not be well explained and so not so easy to understand.

We referred to Teoh et al. defining the outcomes in order to have a stronger comparison between our study and literature findings. We considered technical success each procedure which was safely ended with placement of the drainage or with extraction of the gallbladder.

The Clinical Success was the improvement of the patient within the first week, meaning the resolution of the acute cholecystitis attack.

If the patient represented at the emergency department for another AC attack (this was possible only for PT-GBD and EUS-GBD) it was considered as Recurrence but not due to a treatment failure.

This is true for PT-GBD for which within 6 weeks after the procedure the drain was taken off. For EUS-GBD might be not as clear as in other cases. However, is difficult to define for how long the endoscopic drain can safely and properly solve its function because we do not have enough experience as it is a quite new device.

Overall

PT-GBD

EUS-GBD

LC

OC

p

Recurrence (%)

11 (6.9)

7 (21.9)

3 (14.3)

0 (0.0)

1 (4.0)

<0.001

Recurrence resulted in 3 cases over 21 of EUS-GBD patients…

- What were the criteria to choose whether the cholecystectomy was laparoscopic or open? What is the conversion rate of laparoscopic surgery? Which were initially chosen for the open surgery? In discussion it is commented that it could be due to hemodynamic instability but this is not reflected in the method and I think that is important to clarify this criteria. 

The surgical approach was chosen by the attending surgeon considering the patient physiology, the TG18 score and the AAST score and lastly the laparoscopy expertise (even if in our center all the senior surgeon have a good laparoscopy expertise and if a difficult LC was performed by a surgery resident, a further laparoscopic attempt was done by the senior surgeon before converting to open surgery).

Totally, only 4 patients were converted from LC to OC and they all have an AAST score of more than 3 and a TG18 score of 2 or 3. Furthermore during surgery they badly tolerate pneumoperitoneum and the surgical team finally decide to covert. One of them was a severely ill patient with a frailty score of 8 who failed NOM and was surgically treated. He finally died during intensive Care hospitalization due to his extremely poor physiological status.

- A table specifying the complications that occurred in each group and their classification according to a validated system such as the Clavien Dindo, would help to better understand the manuscript. Probably the authors cannot attach more tables but adding it as supplementary material could be considered if possible. 

Overall

PT-GBD

EUS-GBD

LC

OC

p

Overall complication (%)

41 (25.6)

13 (39.4)

2 (9.5)

13 (16.0)

13 (52.0)

<0.001

Clavien Dindo >3 (%)

16 (10.0)

9 (27.3)

1 (4.8)

2 (2.5)

4 (16.0)

0,001

Clavien Dindo I- V (%)

<0.001

0

126(78.8)

20 (60.6)

20 (95.2)

73 (90.1)

13(52.0)

I

6 (3.8)

4 (12.1)

0 (0.0)

1 (1.2)

1 (4.0)

II

12 (7.5)

0 (0.0)

0 (0.0)

5 (6.2)

7 (28.0)

IIIa

6 (3.8)

5 (15.2)

0 (0.0)

0 (0.0)

1 (4.0)

IIIb

4 (2.5)

3 (9.1)

0 (0.0)

0 (0.0)

1 (4.0)

IVa

2 (1.2)

0 (0.0)

0 (0.0)

0 (0.0)

2 (8.0)

IVb

1 (0.6)

0 (0.0)

1 (4.8)

0 (0.0)

0 (0.0)

V

3 (1.9)

1 (3.0)

0 (0.0)

2 (2.5)

0 (0.0)

We will try to publish this table as supplementary table.

- In the discussion, EUS-GBD is proposed as the first line of treatment, currently it is described as the second line after failure in medical treatment with antibiotics. If it is proposed as first line, in which case do the authors consider it to be an alternative to cholecystectomy? How have the patients been selected to be treated with one or another technique in this study? It is described in methods that have been decided by a multidisciplinary team, but if the authors affirm something as important as changing EUS-GBD to the first line of treatment, I think it would be important to define in which patients a cholecystectomy would be indicated and in which an EUS -GBD.

Patients were selected to undergo EUS-GBD after multidisciplinary evaluation where the fragility of the patients and the expected surgical morbidity/mortality were taken into account. When the NOM treatment failed and the risk for surgery was considered higher ESU-GBD was the procedure of choice If this was not possible due to anatomical factors or endoscopic onsite absence the patient would have undergo a percutaneous drainage.

We will try to explain this flow of events better in the manuscript.

- What happened in patients undergoing cholecystectomy who were not clinically successful? Cholecystectomy usually resolves symptoms and corrects laboratory abnormalities caused by cholecystitis. Can the authors explain why these patients were not clinically successful?

21 patients did not improve after cholecystectomy. Among them are included ones who had complications with a late clinical and chemical response to surgery. If we consider both OC and LC, the overall complication rate was 25% and the overall CS was 80%. The remnant percentage include patients who did not improve without developing a real complication. They were frail and comorbid and they just had a late recovery from surgery. For this reason, they were not considered as early clinical success (within one week from surgery).

In other words, patients did not recover quickly as expected from the sepsis and from correlated organs failure (kidneys, live, lungs), making the recovery longer then a week, meaning they “fail” the quick recover.

- The conclusions talk about using EUS-GBD as a bridge to surgery. Do the authors have experience in cholecystectomy after EUS-GBD? Does prior endoscopic drainage influence in the dificulty of a surgery?

 If the patient has undergone the EUS-GBD, I understand that it was due to high morbidity that contraindicated surgery, I understand that this morbidity will not change after admission, so I understand that the EUS-GBD is proposed as a definitive alternative to surgery given its low rate of adverse events and its low rate of recurrence of acute cholecystitis in patients with high morbidity in whom surgery would not be indicated. Is this what the authors propose?

Literature findings describe the possibility of EUS-GBS as a bridge to surgery. In our experience we cannot agree or disagree because we did not have any cases of surgery performed after EUS-GBD. Since that our conclusion remains theoretical. On the other hand we can assume the safety of this procedure in term of clinical improvement and recurrence. Almost all the cases treated with EUS-GBD were negative for new AC attacks during FU.

It is too early also to define EUS-GBD as a definitive treatment because we need for a longer follow up.we need for a longer follow up.

Reviewer 2 Report

Dear Authors,

Regarding your article “Endosonography-guided versus percutaneous gallbladder drainage versus cholecystectomy in fragile patients with acute cholecystitis. A high-volume center study”, the question of the study is well-defined, fits the journal scope and it’s written in an appropriate way. Data are presented appropriately in tables and figures. Results are well interpreted. The statistical analysis is very well performed. An interesting point to highlight is that anti thrombotic therapy was not suspended for the EUS-GBD and there were not hemorrhagic events.

  • However, there are some issues that need to be addressed: 1) The study is supposed to be conducted in frail patients, but the results of the Clinical Frailty Scoring Scale (CFSS) are not written in Table 1 or anywhere in the text. How frail were these patients and how were they distributed between treatments? 2) In the conclusion it says “Surgery for AC remains the gold standard treatment but this paper evidence the emerging role of EUS-GBD in particular in patients not fit for surgery”. However the ACS of the EUS-GBD group is not high and the frailty score is missing. How do you support this statement? 3) Statistical analysis is based on adverse events and their relationship with ACS. However, it is not stated in the methods section that the main outcome to be considered is the rate of adverse events. 4) The main weakness of this study is the long-term follow-up. Although the follow-up is one year, the recurrence of cholecystitis may be longer term. This has also not been addressed by other studies (1-3 year follow-up). In the discussion this should be noted.
  • Specific comments

INTRODUCTION

- Line 5: polimorbid check spell (polymorbid)

- Line 9: reference of current guidelines is missing (Tokyo Guidelines?)

METHODS

- Line 8-→ TG18: Please state the meaning of TG18 on the text (Tokyo guidelines 2018 ). Check also reference style.

RESULTS?

- TABLE 2: Error on % clinical succes OC (should be 64%)

- TABLE 2: % on recurrence on OC missing: should be 4%

- Page 5: A case of cholangitis after cholecystectomy can not be considered a recurrence of acute cholecystitis, please re-write it.

Looking at the RR, 11 patients returned to the emergency department for a new AC episode. Of these, 7 were from the PT-GBD group, 3 from the EUS-GBD one and one was a case of cholangitis after cholecystectomy.

- Figure 2 PAG 7: meanings of abbreviations missing in figure footnotes

DISCUSSION

- Line 5: From the baseline characteristics analysis, a different distribution amongst frailty and age was outlined; patients treated with PT-GBD or EUS-GBD were older than patients submitted to surgery and had a higher ACS score.

On the methods section you point out that you use the The Clinical Frailty Scale Score (CFSS) to define the fragility level but it is not discussed anywhere on results or discussion section.

- Is there any reason why you decide to choose a treatment instead of another on your patients?

- Last line of discussion: In all cases the device was left in place lifelong without any issues. The main weakness of this study and of the published studies (5,6,27) is the long-term follow-up, with the median of the published studies being one year of follow-up. The recurrence of cholecystitis may be in the longer term and future adverse events have not been taken into account (greater technical difficulties during a future laparoscopic cholecystectomy, morbidity and mortality derived from long-term recurrence, etc.). More studies are required to determine long-term efficacy.

REFERENCES

Review style consistency of references and some errors (words in Italian, some references incomplete eg 14,15,33)

Author Response

Second review answer

Regarding your article “Endosonography-guided versus percutaneous gallbladder drainage versus cholecystectomy in fragile patients with acute cholecystitis. A high-volume center study”, the question of the study is well-defined, fits the journal scope and it’s written in an appropriateway. Data are presented appropriately in tables and figures. Results are well interpreted. The statistical analysis is very well performed. An interesting point to highlight is that anti thrombotictherapy was not suspended for the EUS-GBD and there were not hemorrhagic events. 

However, there are some issues that need to be addressed: 1) The study is supposed to be conducted in frail patients, but the results of the Clinical Frailty Scoring Scale (CFSS) are not written in Table 1 or anywhere in the text. How frail were these patients and how were they distributed between treatments? 

We coincide with this missing that we are going to insert in the final table 1.

The CFSS distribution was different among groups with statistical significance. As expected CFSS was higher in the PT-GBD group.The difference highlight, in our opinion, a good evaluation and choice. In fact these patients would have been expected to have a poor surgery response due to their frailty level. It must be noticed that patients undergoing EUS-GBD had a 2-point-lower CFSS. This is probably due to the retrospective nature of the current study and the short-time experience in EUS-GBD field. As previously explained AC patients candidate to an invasive procedure were evaluated by surgeon, endoscopist and radiologist and frail patients should have been avoided from surgery and addressed to PT vs EUS guided drainage. However, the short-time experience could have compelled the multi-disciplinary team to avoid EUS-GBD if not completely safe.

Overall

PT-GBD

EUS-GBD

LC

OC

p

Clinical Fraility Scale (median [range])

3.00 [1.00, 8.00]

6.00 [2.00, 8.00]

4.00 [2.00, 8.00]

3.00 [1.00, 8.00]

3.00 [2.00, 7.00]

<0.001

 2) In the conclusion it says “Surgery for AC remains the gold standard treatment but this paper evidence the emerging role of EUS-GBD in particular in patients not fit for surgery”. However, the ACS of the EUS-GBD group is not high and the frailty score is missing. How do you support this statement? 

The ACS score for EUS-GBD patients is lower than the score of PT-GBD because calculating the score the “weight” of the procedure is higher than the physiological and pathological values’ “weight”. If we look at a standard patient undergoing a cholecystostomy procedure he would have a higher score than one undergoing surgery or a EUS-GBD placement. This is an inner problem of the calculator that could be due to the fact it is based on a real cases database (in which PT-GBD patients have poorer outcome).

We agree on the fact that CFSS could be a good score to associate to better describe our population and surely, we are going to discuss it in the revised version of the article.

It is true that PT-GBD have a higher CFSS, and we have already commented this aspects.

It must be noticed that EUS-GBD patients have a median fragility level that was “mild” but higher than the surgery group value. From these results we could assume that EUS-GBD is a good choice for mild frail patient (this is supported by our finding on RR, AEs and SAEs rate) but it should be further evaluated in highly frail patients to better support its safety and feasibility.

3) Statistical analysis is based on adverse events and their relationship with ACS. However, it is not stated in the methods section that the main outcome to be considered is the rate of adverse events. 

We will properly modify our methods section to clarify this aspect.

4) The main weakness of this study is the long-term follow-up. Although the follow-up is one year, the recurrence of cholecystitis may be longer term. This has also not been addressed by other studies (1-3 year follow-up). In the discussion this should be noted.

We concur on the short follow up issue, and we will clarify this aspect in the discussion.

Specific comments

INTRODUCTION

- Line 5: polimorbid check spell (polymorbid)

- Line 9: reference of current guidelines is missing (Tokyo Guidelines?)

METHODS

- Line 8-→ TG18: Please state the meaning of TG18 on the text (Tokyo guidelines 2018). Check also reference style. 

RESULTS

- TABLE 2: Error on % clinical succes OC (should be 64%)

- TABLE 2: % on recurrence on OC missing: should be 4%

- Page 5: A case of cholangitis after cholecystectomy can not be considered a recurrence of acute cholecystitis, please re-write it. 

Looking at the RR, 11 patients returned to the emergency department for a new AC episode. Of these, 7 were from the PT-GBD group, 3 from the EUS-GBD one and one was a case of cholangitis after cholecystectomy.

- Figure 2 PAG 7: meanings of abbreviations missing in figure footnotes

DISCUSSION

- Line 5: From the baseline characteristics analysis, a different distribution amongst frailty and age was outlined; patients treated with PT-GBD or EUS-GBD were older than patients submitted to surgery and had a higher ACS score.

On the methods section you point out that you use the The Clinical Frailty Scale Score (CFSS) to define the fragility level but it is not discussed anywhere on results or discussion section. 

- Is there any reason why you decide to choose a treatment instead of another on your patients?

- Last line of discussion: In all cases the device was left in place lifelong without any issues. The main weakness of this study and of the published studies (5,6,27) is the long-term follow-up, with the median of the published studies being one year of follow-up. The recurrence of cholecystitis may be in the longer term and future adverse events have not been taken into account (greater technical difficulties during a future laparoscopic cholecystectomy, morbidity and mortality derived from long-term recurrence, etc.). More studies are required to determine long-term efficacy.

REFERENCES 

Review style consistency of references and some errors (words in Italian, some references incomplete eg 14,15,33)
